# Development modeling of *Lucilia sericata* (Diptera: Calliphoridae)

Amanda Roe[1] and Leon G. Higley[2]

[1] Biology Program, College of Saint Mary, Omaha, NE, United States
[2] School of Natural Resources, University of Nebraska-Lincoln, Lincoln, NE, United States

## ABSTRACT

The relationship between insect development and temperature has been well established and has a wide range of uses, including the use of blow flies for postmortem (PMI) interval estimations in death investigations. To use insects in estimating PMI, we must be able to determine the insect age at the time of discovery and backtrack to time of oviposition. Unfortunately, existing development models of forensically important insects are only linear approximations and do not take into account the curvilinear properties experienced at extreme temperatures. A series of experiments were conducted with *Lucilia sericata*, a forensically important blow fly species, that met the requirements needed to create statistically valid development models. Experiments were conducted over 11 temperatures (7.5 to 32.5 °C, at 2.5 °C) with a 16:8 L:D cycle. Experimental units contained 20 eggs, 10 g beef liver, and 2.5 cm of pine shavings. Each life stage (egg to adult) had five sampling times. Each sampling time was replicated four times, for a total of 20 measurements per life stage. For each sampling time, the cups were pulled from the chambers and the stage of each maggot was documented morphologically through posterior spiracle slits and cephalopharyngeal skeletal development. Data were normally distributed with the later larval stages (L3f, L3m) having the most variation within and transitioning between stages. The biological minimum was between 7.5 °C and 10 °C, with little egg development and no egg emergence at 7.5 °C. Temperature-induced mortality was highest from 10.0 to 17.5 °C and 32.5 °C. The development data generated illustrates the advantages of large datasets in modeling *Lucilia sericata* development and the need for curvilinear models in describing development at environmental temperatures near the biological minima and maxima.

## INTRODUCTION

In the past, insect development research has focused on agriculture pests and disease vectors, with temporal accuracy levels of days (or weeks) considered acceptable since the focus was on economic thresholds and vector prevention (*Higley & Haskell, 2010*). In the last 50 years, however, there has been a growing interest in the development of necrophagous insects. Unfortunately, unlike agricultural and medically important insects, whose biology had been studied down to eye color during pupal stages and mode of infection from digestive track to mouthparts, necrophagous flies had no such

Corresponding author
Amanda Roe, aroe@csm.edu

fervor surrounding them and their development data was relegated to a few ecological studies (e.g., *Mackerras, 1933*; *Fuller, 1934*; *Kamal, 1958*). Interest in necrophagous insects, specifically blow flies (Diptera: Calliphoridae) continued (and continues) to rise with the (re)discovered usefulness of their development for postmortem interval estimations (PMI or time since death) (*Greenberg, 1991*).

The blow fly, *Lucilia sericata*, is a species among that group of necrophagous insects. They are ubiquitous, covering a broad range of landscapes, and may be one of the most common blow fly species in the world (*Hall, 1948*; *Greenberg, 1991*; *Byrd & Castner, 2010*). As such, this species is at the forefront of biological and development studies because of its role in maggot therapy, animal (including human) myiasis, and postmortem interval estimations.

When found on a human body, the developing eggs, larvae, or pupae of *L. sericata* can be used as an index pointing to the postmortem interval (PMI). Estimating the PMI is crucial in most human death investigations because time since death is needed to properly reconstruct events before and after death. Using development in estimating PMI is dependent on determining the insect age at the time of discovery and backtracking to time of oviposition. Consequently, understanding temperature-specific development rates is essential, yet development rate concepts from agricultural pest development data sets/models are being applied to postmortem interval estimations, implying levels of precision greater than the data allow (e.g., see discussion in *Higley & Haskell, 2010*). Among the most substantial studies on *Lucilia sericata* development (in terms of temperatures examined and overall citations) are those of *Kamal (1958)*, *Ash & Greenberg (1975)*, *Greenberg (1991)*, *Anderson (2000)* and *Grassberger & Reiter (2001)* (Table 5). However, many methodological problems exist in these studies relative to determining development rates, including insufficient replication, inconsistent temperature ranges or too few temperatures, no indication of temperature variability, non-life stage specific results, and unspecified or inconsistent sampling intervals. In total, limitations with existing data make it difficult to apply error rates or confidence intervals. These are key problems, not only in *L. sericata* data, but in most blow fly developmental data. Generally, there is little consistency between studies making it difficult to pool data or make comparisons.

Although various procedures exist for measuring development rates, the simplest and most common is regression (either linear or curvilinear). However, data for use in regressions must meet specific criteria (*Snedecor & Cochran, 1989*). Independent variables (temperatures in development regressions) must be equally spaced, otherwise values at the ends of the examined range have a disproportionate influence on the relationship. Additionally, independent variables are assumed to have zero or negligible variation, otherwise systematic error can occur in the calculated relationship.

Perhaps the most cited flaw in regression analyses is to fit a linear model to curvilinear data (*Snedecor & Cochran, 1989*). This issue is especially pertinent for blow fly development, because degree-day models are based on linear regression, yet it is well established that development is curvilinear (*Higley, Pedigo & Ostlie, 1986*; *Higley & Haskell, 2010*).

The solution is to explicitly limit the temperature range for which a degree-day model is valid to linear portions of the development curve (*Nabity, Higley & Heng-Moss, 2007*).

Another pertinent methodological point is how to replicate temperature treatments. By definition, an experimental unit is the thing to which a treatment is applied, and experimental units must have independent replication. Because temperatures are applied in growth chambers, the chamber is (by definition) the experimental unit (*Snedecor & Cochran, 1989*). Measurements of within growth chamber temperature variations and systematic differences between experienced and nominal chamber temperatures (*Nabity, Higley & Heng-Moss, 2007*) demonstrate that chamber replication and/or temperature measurements within chamber locations are necessary to provide proper estimates of experimental error and to avoid systematic measurement errors. *Richards & Villet (2008)* discuss how deficiencies in development data can reduce the accuracy of PMI estimations. In particular, sampling errors and models based on too few temperatures directly impact statistical validity and error rates. Besides the obvious reason of unknown or high error rates making it difficult to reach conclusions, known error rates are a requirement for Federal Rules of Evidence (Rule 702: Testimony by Expert Witnesses) under the *Daubert standard*. Judges can use the "the known or potential rate of error of the technique or theory when applied" as an assessment of reliability of the evidence being presented.

Many of the issues associated with current data sets likely relates to the sheer amount of resources required to establish a complete, statistically valid development model. Preparatory work averages between 15 and 18 h per temperature. These hours include cutting weighed liver, labeling and organizing experimental units, counting eggs, and putting all units together before they go in the chambers. After set up, the hours required for actual sampling can easily exceed 120 h (at an average of one hour per sampling time). A considerable time (not included in the hours above) is required for colony maintenance. Multiple colonies are required to for high egg production and to prevent time loss from loss of a colony.

Thus, although the experiments are time and labor intensive, we conducted a series of experiments that cover a broad range of temperatures, have large sample sizes, and have the consistent sampling times required to create a statistically valid development model.

## METHODS AND MATERIALS

### Flies

*Lucilia sericata* were obtained from colonies maintained at the University of Nebraska-Lincoln (Lincoln, Nebraska). Colonies were established in October 2010, from field-collected insects from Morgantown, West Virginia. At research time, the colonies had achieved 100 generations without addition of new flies to reduce genetic variation within the colony. Adult flies were maintained in screen cages (46 cm × 46 cm × 46 cm) (Bioquip Products, California) in a rearing room at 27.5 °C (±3 °C), with a 16:8 (L:D) photoperiod. Multiple generations were maintained in a single cage, and ca. 1,000 adult flies were introduced every 1–2 weeks. Adults had access to granulated sugar and water *ad libitum*, and raw beef liver for protein and as an ovipositional substrate. After egg laying, eggs and

liver were placed in an 89 ml plastic cup, which was surrounded by pine shavings in a 1.7 L plastic box. The pine shavings served as a pupation substrate. The 1.7 L box was placed in a I30-BLL Percival biological incubator (Percival Scientific, Inc., Perry, Iowa, USA) set at 26 °C (±1.5 °C). After eclosion, adults were released into the screened cages.

## Incubators

Incubator information has been previously discussed in *Lein (2013)*. Pertinent information has been revisited here. Incubators were customized model SMY04-1 DigiTherm® CirKinetics Incubators (TriTech Research, Inc., Los Angeles, California, USA). The DigiTherm® CirKinetics Incubator have microprocessor controlled temperature regulation, internal lighting, recirculating air system (to help maintain humidity), and use a thermoelectric heat pump (rather than coolant and condenser as is typical with larger incubators and growth chambers). Customizations included the addition of a data port, vertical lighting (so all shelves were illuminated), and an additional internal fan. The manufacturer's specifications indicate an operational range of 10–60 °C ± 0.1 °C. It is worth noting that a range of ±0.1 °C is an order of magnitude more precise than is possible in conventional growth chambers. Although growth chambers have been shown to display substantial differences between programmed temperatures and actual internal temperatures (*Nabity, Higley & Heng-Moss, 2007*), we tested the customized DigiTherm® CirKinetics incubators in a replicated study and found internal temperatures on all shelves within incubators did not vary more than 0.4 °C from the programmed temperature (data not shown). Given this measured accuracy, the incubators could be used without the need for additional internal temperature measurements or risk of systematic error (as there was <0.4 °C internal variation from the programmed temperature).

## Experimental Design

The study comprised eleven temperatures (7.5, 10, 12.5, 15, 17.5, 20, 22.5, 25, 27.5, 30, and 32.5 °C) with a light:dark cycle of 16:8. Twenty eggs (collected within 30 min of oviposition) were counted onto a moist black filter paper triangle and placed in direct contact with 10 g of beef liver in a 29.5 mL plastic cup. The cup was placed in a 7 cm × 7 cm × 10 cm plastic container that had 2.5 cm of wood shavings in the bottom. The container was then placed randomly in an incubator. There were 27 container locations within each chamber (9 locations per shelf). Containers were randomized by chamber using a random number generator in Excel (Microsoft Excel 2007). Each life stage (egg–1st stage, 1st–2nd stage, 2nd–3rd stage, 3rd–3rd migratory, 3rd migratory-pupation, pupation-adult) was calculated using *Kamal*'s (*1958*) data, which was converted to accumulated degree hours (ADH) and divided equally into five sampling times (Table 1). Each sample was replicated four times, with a total of 20 samples per life stage. During each sample time, a container was pulled from each of the four incubators and the stage of each maggot was documented morphologically using the posterior spiracular slits and cephalopharyngeal skeleton.

During egg hatch, a larva was recorded as 1st stage if they had broken the egg chorion and were actively emerging. Pharate larvae (larvae that have undergone apolysis but not

**Table 1** *Lucillia sericata* **sample times (hours after oviposition).** Sample times for *Lucilia sericata* were calculated by converting the minimum and maximum data reported in *Kamal (1958)* into accumulated degree hours (ADH). The ADH's were calculated for each life stage and sampling temperature, converted back into hours and divided into 5 equal sample times.

| Life stage | Temperature °C | | | | | | | | | | |
|---|---|---|---|---|---|---|---|---|---|---|---|
| | 7.5 | 10.0 | 12.5 | 15.0 | 17.5 | 20.0 | 22.5 | 25.0 | 27.5 | 30.0 | 32.5 |
| Egg–1st | 35 | 35 | 35 | 17 | 12 | 9 | 7 | 6 | 5 | 4 | 4 |
| 1st–2nd | 56 | 56 | 56 | 28 | 19 | 14 | 11 | 9 | 8 | 7 | 6 |
| 2nd–3f | 79 | 79 | 79 | 39 | 26 | 20 | 16 | 13 | 11 | 10 | 9 |
| 3f–3m | 143 | 143 | 143 | 71 | 48 | 36 | 29 | 24 | 20 | 18 | 16 |
| 3m–Pupal | 335 | 335 | 335 | 167 | 112 | 84 | 67 | 56 | 48 | 42 | 37 |
| Pupal–Adult | 527 | 527 | 527 | 263 | 176 | 132 | 105 | 88 | 75 | 66 | 59 |

ecdysis) were recorded as the earlier stage (e.g., 3rd stage spiracular slits can be seen beneath the current spiracular slits would be recorded as 2nd stage), since they had not yet molted. Larvae were considered 3rd migratory when they stopped feeding, left the liver, and began burrowing through the pine shavings. Pupariation began when larvae had a shortened body length and no longer projected mouth hooks when put in the larval fixative KAAD (kerosene-acetic acid-dioxane). There were times when a larva appeared to be entering the puparium stage but would extend its body length and begin crawling if disturbed or placed in KAAD. These larvae were recorded as 3rd migratory. All life stages were preserved in 70% ethyl alcohol. Third and 3rd-migratory stages were fixed in KAAD for 48 h and transferred to 70% ethyl alcohol.

## Analysis

In the literature, the time in a stage is typically reported as a single number. Because variation exists among individuals in their development times, it is essential to use an appropriate indicator reflecting when individuals transition from one stage to another. With normally distributed variation this estimator is a mean, but with alternative distributions other measures are more appropriate. Consequently, the distribution of individuals during stage transitions must be determined. Details on stage transition modeling and its importance in properly determining development rates are reported elsewhere (*Roe, 2014*). Here, we summarize the procedures used in determining time in stage.

First, we modeled stage transitions using TableCurve 2d, version 5.01 (SYSTAT Software Inc, San Jose, California, http://www.sigmaplot.com/products/tablecurve2d/ Tablecurve2d.php), and Prism, version 6.02 (GraphPad Software, Inc., La Jolla, California, http://www.graphpad.com/scientific-software/prism/). We fit one of four Gaussian functions (specifically, a regressed proportion (percentage) in stage versus time, at each temperature tested. Then, stage durations were determined, probit models were constructed, and the 50% transition point was determined.

For all regression analyses, the data were examined closely to determine their propriety for inclusion in analysis. In a few instances, individuals were sampled with extraordinarily

extended durations. These were treated as outliers and excluded from analysis. Details on all data used are included in Appendix S1.

## DEGREE DAYS

Degree day requirements were calculated with a combination of regression analyses and iterative analyses to ensure the resulting degree day models reflected only the linear portion of the insect development curve. The outline of these procedures is:

1. Determine the stage transitions by fitting Gaussian curves to the proportion of insects entering the new stage vs. time for each temperature (curves were calculated for L1, L2, L2f, L3m, P, and A). Only data for the first portion of each curve (0%–100%) was included in the regression which reflects the stage transition.

2. Calculate the 50% transition point from the Gaussian curve for each stage and temperature combination.

3. With data from 2, determine time in stage by subtraction between 50% transition points.

4. Express development times in days (rather than hours, as data was initially determined) and calculate 1/days for each time to transition and stage duration.

5. By linear regression, estimate the relationship between development rate (1/days to transition or stage) vs. temperature) to determine the slope and x-intercept. Each resulting regression was runs tested to identify non-linearity, and where nonlinearity was indicated, points were excluded from the regression until any non-linearity was eliminated. (Runs testing is a procedure by which iterative calculations are used to distinguish linear from non-linear points in a regress (*Mutulsky, 1995*; *GraphPad Software, Inc., 2014*)). Primarily, non-linearity was associated with low and high temperatures (as expected) and indicated in development graphs. The regression of 1/days vs. temperature is conventional in degree day determination, but the use of runs testing to identify non-linear points in the regression has not been. To the best of our knowledge this approach was first used in *Nabity, Higley & Heng-Moss (2007)* to ensure that assumptions underlying degree day analysis were met.

6. From the resulting linear regressions, the x-intercept represents the developmental minimum and 1/slope represents the accumulated degree days required for an event (stage transition or stage duration) (*Arnold, 1959*). Although this point usually represents the end of most degree day determinations, it is still possible at this point to have included data in the linear regressions that are not properly part of the linear portion of the development curve. Consequently, we did additional calculations and corrections to determine the validity of our degree day models.

Using regression results we calculated degree day accumulations for each experimentally determined combination of temperature and time of transition or stage duration. We then did a linear regression of these data and evaluated the resulting lines for linearity and slope. To meet the core assumption of degree day models, a regression of degree day

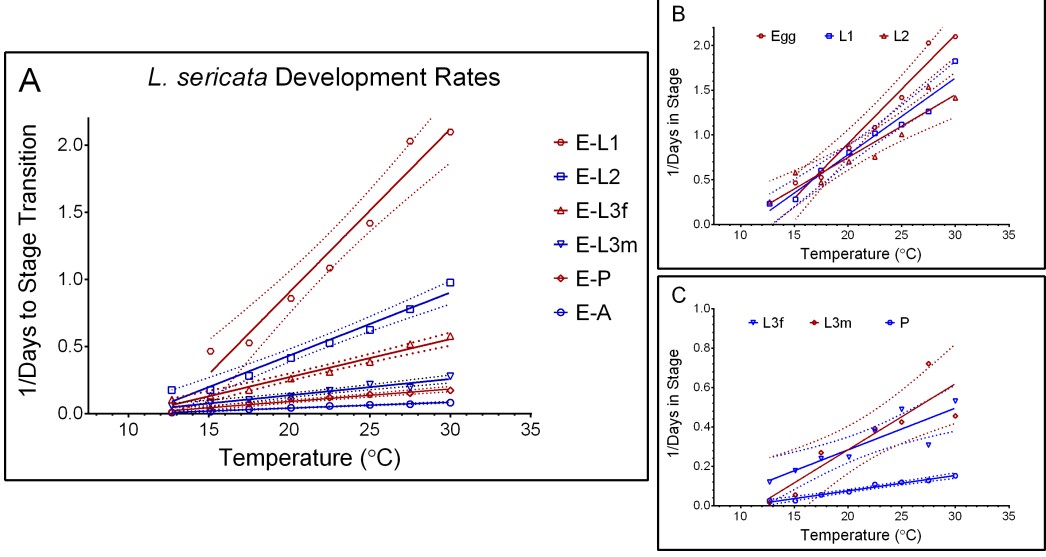

**Figure 1  Development rates of *Lucilia sericata* by stage.** From 10.0 to 32.5 °C, with 95% confidence intervals represented by dotted lines, Life stages egg-L2 are indicated in 1B and stages L3-P in 1C.

accumulations must be linear and have no slope. Where our results did not meet these requirements, we removed points (again, at high and low temperatures), and recalculated both the 1/days regression and the accumulated degree days regressions (steps 5–7). We repeated this process until we arrived at linear relationships meeting all degree day assumptions, and noted the range of temperatures for which the resulting equation was valid.

## RESULTS AND DISCUSSION

All calculations can be found in Appendix S1.

We observed substantial variation in stage transition times (*Roe, 2014*) and stage durations comparable to that reported by *Tarone & Foran (2006)*. The largest variation was observed during the L3m and pupation stages, regardless of temperature, with the variation largest at 10.0 °C through 17.5 °C (L3m and pupation) and 32.5 °C (L3m) (Fig. 1). These stages are also the longest life stages (by proportion) (Table 2).

With minimal egg development and no egg eclosion at 7.5 °C, no data were reported. There is evidence, however, that the biological developmental minimum for *L. sericata* is between 7.5 °C and 10.0 °C, since there were individuals that successfully emerged as adults at 10.0 °C. Although the adults at 10.0 °C and 12.5 °C were normal-sized, the total number of individuals that survived into adulthood was very small compared to the other temperatures. High mortality rates and reduced developmental rates have been reported at 35.0 °C (*Ash & Greenberg, 1975*), indicating suboptimum temperatures on both ends of the spectrum can impact survivorship and growth. Since extreme temperatures lead to extreme biological variation, there is a disruption to normal gene expression, which can alter the hormones and proteins needed in molting and maintenance. The biological variation at these temperatures may be an inherent variation in *L. sericata* that allows the

**Peer**J

**Table 2** **Percent of time *Lucilia sericata* in stage.**

| Temp | % Time in stage | | | | | |
|---|---|---|---|---|---|---|
| | Egg | L1 | L2 | L3f | L3m | P |
| 10.4 | 4.7% | 5.4% | 1.4% | 5.5% | 37.3% | 45.7% |
| 12.7 | 1.4% | 3.3% | 3.3% | 6.9% | 51.9% | 33.2% |
| 15.1 | 3.0% | 4.6% | 2.8% | 7.9% | 52.2% | 29.5% |
| 17.5 | 6.2% | 4.2% | 7.5% | 13.2% | 11.7% | 57.1% |
| 20.1 | 5.1% | 5.6% | 6.0% | 18.2% | 4.3% | 60.8% |
| 22.5 | 5.0% | 5.9% | 7.4% | 14.9% | 14.5% | 52.3% |
| 25.0 | 4.5% | 5.2% | 7.1% | 13.2% | 15.2% | 54.7% |
| 27.5 | 5.2% | 3.7% | 5.4% | 22.0% | 9.7% | 54.1% |
| 30.0 | 3.9% | 4.1% | 6.0% | 14.7% | 18.1% | 53.2% |
| 32.5 | 3.9% | 3.0% | 3.4% | 16.7% | 24.1% | 48.9% |
| **Mean** | 4.3% | 4.5% | 5.0% | 13.3% | 23.9% | 49.0% |

species to survive in suboptimal conditions and also successfully maintain populations throughout the world.

Based on temperatures measured and methodology, the previous study most directly comparable with our data was that of *Kamal (1958)*. In the early life stages (E to L2), the percent in stage observed here was within 2.1% of that reported by *Kamal (1958)* (Table 4). In the later life stages there was a difference of 10.5% (L3f) to 16.2% (L3m) (Table 4). Although we might expect greater variation at later life stages, differences in development times might also reflect differences in methods. *Kamal (1958)* used constant lighting during his experiments, and *Nabity, Higley & Heng-Moss (2007)* showed significant delays in developmental under constant light compared to 16:8 L:D cycles. However, the development times are comparable, indicating that geographic variation in development may be less than the inherent variation in development of *L. sericata.*

In comparison with other previous work (besides Kamal), differences among studies seem likely to be associated with methodology (see also *Tarone & Foran, 2006*). We say this because sampling times used to determine stage duration were not consistent among previous studies and in some instances may have been inappropriate (Table 5). For example, sampling extensively at the start of a stage would shift the developmental distribution to the left, skewing the mean. Alternately, sampling the largest individuals (a common practice in forensic entomology casework) changes the age cohort data by using the maxima and treating them as normal data points, not outliers (*Richards & Villet, 2008*).

Our data indicate that stage transitions are integral to generating realistic, accurate development data (*Roe, 2014*). Transitions occur over a period of hours to days and it is difficult to discuss development without knowing the transition periods between stages, since the vast majority of time in stage is spent as a mixed-age population. Most error occurs in two areas: temperature and stage/age calculations. Being aware of transition times and having consistently spaced sampling times and temperatures lets us attach error margins to the data, reducing the error in the age/stage calculations. By reducing error in

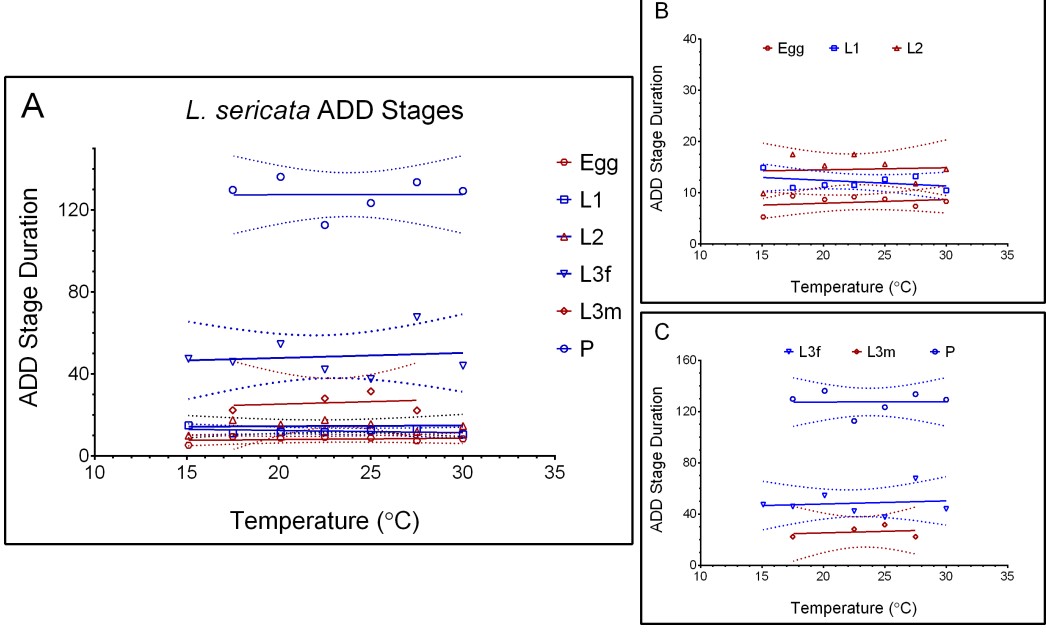

**Figure 2 Accumulated degree day stage durations of *Lucilia sericata*.** From 10.0 to 32.5 °C, with 95% confidence intervals represented by dotted lines. Life stages egg-L2 are indicated in 2B and stages L3-P in 2C.

one of two areas, we accomplish two things: we can focus attention on the error associated with temperature (fluctuating versus constant, unknown versus close by, etc.) and we can attach known error rates to PMI estimates.

Currently, the data sets available make attaching error rates difficult or impossible, depending on the data presentation. While these data let us generate error rates, they are only accurate for the linear portion of the life stages (or the temperature range that the ADD are valid). The assumption of linearity was met for all life stages, but not all temperatures (Fig. 2). Not surprisingly, L3m has the shortest linear temperature range from 17.5 to 30.0 °C. L1 also has a shortened range from 12.5 to 32.5 °C (Table 3).

In conventional uses of degree days (e.g., *Arnold, 1959*), using multiple methods to ensure only linear development data are used in determining degree-day models is not undertaken. Presumably this omission has occurred because it is well recognized that degree-day models use assumptions of linearity to describe what is known to be a curvilinear relationship, so approaches for improving accuracy have focused on curvilinear model development (*Wagner et al., 1984*; *Higley & Haskell, 2010*) rather than on improving linear degree day accuracy. Additionally, most conventional uses of degree days with insects involve modeling population level phenomena, where other sources of error (particularly in temperature data) and resolution (of days) are such that more precision in how degree-day models are developed may not be warranted. In contrast, with forensic use of degree-day models, the potential inaccuracy associated with including non-linear data in the calculation model introduces systematic error that could easily be significant in using degree days for estimating postmortem intervals. As a practical matter, systematic

**Table 3  Linear regression results.** Linear regression results (from Graph Pad Prism) of *Lucilia sericata*, with excluded (non-linear) points indicated by an empty cell. Accumulated Degree Days (ADD) were indicated by the slope of the regression line, and the developmental minimum was indicated by the x-intercept value (*Arnold, 1959*). For comparison, regression-based ADD were compared to mean ADD calculated across temperatures. The range of the linear regression indicates the temperature limits at which the assumption of linearity between temperature and development is valid.

| Temperature | Transition ADD by 1/Day | | | | | | Stage ADD by 1/Days | | | | | |
|---|---|---|---|---|---|---|---|---|---|---|---|---|
| | E–L1 | E–L2 | E–L3f | E–L3m | E–P | E–A | Egg | L1 | L2 | L3f | L3m | P |
| 10.4 | | | | | | | | | | | | |
| 12.7 | | | | | | | | | | | | |
| 15.1 | 5.3 | 24.4 | 34.4 | 82.4 | | | 12.0 | 14.9 | 9.9 | | | |
| 17.5 | 9.4 | 24.0 | 40.3 | 86.7 | 97.8 | 218.2 | 15.3 | 11.0 | 17.5 | 45.8 | 22.3 | 129.8 |
| 20.1 | 8.7 | 22.4 | 36.9 | 89.3 | 88.0 | 217.3 | 12.4 | 11.5 | 15.3 | 54.5 | | 136.2 |
| 22.5 | 9.2 | 22.4 | 39.0 | 81.1 | 102.9 | 210.3 | 12.1 | 11.5 | 17.5 | 42.2 | 28.1 | 112.7 |
| 25.0 | 8.8 | 22.7 | 37.7 | 75.3 | 102.4 | 221.2 | 10.9 | 12.6 | 15.6 | 37.5 | 31.6 | 123.4 |
| 27.5 | 7.4 | 21.5 | 33.1 | 97.2 | 112.8 | 242.1 | 8.9 | 13.2 | 11.8 | 67.7 | 22.2 | 133.6 |
| 30.0 | 8.3 | 19.7 | 33.9 | 76.8 | 114.1 | 239.7 | 9.8 | 10.5 | 14.6 | 44.0 | 40.5 | 129.3 |
| 32.5 | | | | | | | | | | | | |
| | | | | | | **Linear regression results** | | | | | | |
| **Dev. Min (x-intercept):** | 12.6 | 10.8 | 10.5 | 8.8 | 10.3 | 10.7 | 9.5 | 10.9 | 9.3 | 6.6 | 11.5 | 10.4 |
| **Regression ADD (1/slope):** | 8.2 | 21.3 | 35.2 | 82.5 | 107.5 | 230.2 | 10.3 | 11.7 | 14.3 | 47.2 | 29.8 | 127.9 |
| **$r^2$:** | 0.96 | 0.97 | 0.97 | 0.95 | 0.96 | 0.98 | 0.96 | 0.96 | 0.89 | 0.78 | 0.80 | 0.97 |
| **n** | 4 | 4 | 4 | 4 | 4 | 4 | 3 | 3 | 4 | 4 | 4 | 4 |
| **Range min:** | 15 | 15 | 15 | 15 | 17.5 | 17.5 | 15 | 15 | 15 | 17.5 | 17.5 | 17.5 |
| **Range max:** | 30 | 30 | 30 | 30 | 30 | 30 | 30 | 30 | 30 | 30 | 30 | 30 |
| **Calculated ADD mean** | 8.2 | 22.5 | 36.5 | 84.1 | 103.0 | 224.8 | 11.6 | 12.2 | 14.6 | 48.6 | 28.9 | 127.5 |
| **SE** | 1.31 | 1.46 | 2.56 | 7.05 | 8.86 | 11.86 | 1.92 | 1.41 | 2.61 | 9.95 | 6.80 | 7.70 |
| **% deviation (calc vs. regression ADD)** | −0.8% | 5.4% | 3.5% | 1.9% | −4.2% | −2.3% | 13.3% | 4.1% | 2.4% | 3.1% | −3.0% | −0.3% |
| **ADD range min:** | 15 | 15 | 15 | 15 | 17.5 | 17.5 | 15 | 15 | 15 | 17.5 | 17.5 | 17.5 |
| **ADD range max:** | 30 | 30 | 30 | 30 | 30 | 30 | 30 | 30 | 30 | 30 | 30 | 30 |

**Table 4  Comparison between *Kamal (1958)* and *Roe (2014)* of *Lucilia sericata* as percent in stage.**

| Source | Temp | Egg | L1 | L2 | L3f | L3m | P |
|---|---|---|---|---|---|---|---|
| **This study** | 27.5 | 5.2% | 3.7% | 5.4% | 22.0% | 9.7% | 54.1% |
| **Kamal** | 26.7 | 5.2% | 5.7% | 3.4% | 11.5% | 25.9% | 48.0% |
| **Difference** | | 0.0% | −2.1% | 1.9% | 10.5% | −16.2% | 5.8% |

error will result in under or overestimates in the PMI, depending on temperatures errors not of hours but of days.

Looking past the specifics reported here on *L. sericata*, the most far-reaching implication of this work is the recognition that existing data on the development of forensically important insects may not be comprehensive enough for precision in PMI estimates. Irrespective of the type of developmental modeling used, whether linear (degree day) or curvilinear, the ability to make estimates of insect development at a forensically necessary

Roe and Higley (2015), *PeerJ*, DOI 10.7717/peerj.803

**Table 5** Methods comparison of five development papers for *Lucilia sericata.*

| Reference | Locality | Temperatures | Analysis | Larval diet | Stages | L:D cycle | Replications | Total maggots/sample | Sample times |
|---|---|---|---|---|---|---|---|---|---|
| *Kamal (1958)* | Colorado, U.S. | 27.6 | Mode | Beef liver | E, L1, L2,L3f, L3m, P | Constant | Undefined | Undefined | Undefined |
| *Ash & Greenberg (1975)* | Illinois, U.S. | 19, 27, 35 | Mean | Macerated liver | E, Larval, P | Undefined | Undefined | Undefined | Undefined |
| *Greenberg (1991)* | Illinois, U.S. | 19, 22, 29, 35 | Minimum | Ground beef | E, L1, L2, L3f, L3m, P | Undefined | Undefined | Undefined | Undefined |
| *Anderson (2000)* | British Columbia, Canada | 15.8, 20.7, 23.3 | Minimum and Maximum | Beef liver | E, L1, L2, L3f, L3m, P | Undefined | 8, 9, 2 | 20, returned to jar | Eggs-1 to 2 h L1, L2- 3 to 4 times/day Later stages-2 to 3 times/day |
| *Grassberger & Reiter (2001)* | Vienna, Austria | 15, 17, 19, 20, 21, 22, 25, 28, 30, 34 | Minimum | Beef liver | E, L1, L2, L3f, L3m, P | Presumed 12:12 | 10/temp | 4 | Every 4 h After peak feeding-every 6 h |

resolution of hours or 1–2 days, requires better estimates of stage transitions and data sets than currently exist. Additionally, the high levels of variation we observed in development further illustrate the crucial need for proper replication in developmental work, if we are going to be able to make scientifically valid statements on variation. Consequently, despite the time, difficulty, and expense of comprehensive developmental studies, our results indicate such data for all forensically important blow flies are essential to meet the need for accurate PMI estimates based on insect development.

## ACKNOWLEDGEMENTS

We thank Dr. Jeff Wells and Dr. Anne Perez for the initial *L. sericata* used to start our colonies and Christian Elowsky for his assistance in the lab.

The opinions, findings, and conclusions or recommendations expressed in this publication/program/exhibition are those of the author(s) and do not necessarily reflect those of the Department of Justice. NIJ defines publications as any planned, written, visual or sound material substantively based on the project, formally prepared by the award recipient for dissemination to the public.

### Funding

Financial support for this research was provided by the National Institute of Justice, Office of Justice Programs, U.S. Department of Justice (Grant No. 2010-DN-BX-K231). The funders had no role in study design, data collection and analysis, decision to publish, or preparation of the manuscript.

### Grant Disclosures

The following grant information was disclosed by the authors:
National Institute of Justice, Office of Justice Programs, U.S. Department of Justice: 2010-DN-BX-K231.

### Competing Interests

Author Leon Higley is an Academic Editor for Peer J.

### Author Contributions

- Amanda Roe conceived and designed the experiments, performed the experiments, analyzed the data, wrote the paper, prepared figures and/or tables, reviewed drafts of the paper.
- Leon G. Higley conceived and designed the experiments, analyzed the data, contributed reagents/materials/analysis tools, wrote the paper, prepared figures and/or tables, reviewed drafts of the paper.

### Supplemental Information

Supplemental information for this article can be found online at http://dx.doi.org/10.7717/peerj.803#supplemental-information.

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
