# Peer review of "Development modeling of Lucilia sericata (Diptera: Calliphoridae)"

_PeerJ, doi:10.7717/peerj.803_

## Round 0.1 · original submission · Major Revisions

Please follow closely the comments by the two reviewers.

Reviewer 1 ·

Basic reporting

Even though the current study presented by Roe and Highley is properly done and documented, the overall organization of the text is not adequate. The 'Methods and Materials' section should focus only on the tools they use, and the comparisons with other experimental approaches/models should be done in the discussion section. Results are mainly described in the discussion section instead of the 'Results' section itself. Some figures are not referred in the text or are mistakingly labelled. Tables are not described in the Results section, only in the Discussion.
Other minor notes: The authors mention that in the last 50 years 'there has been a growing interest in the development of necrophagous insects', but their references are much older than 50 years. Some phrases should be clarified (e. g. p10, line 216). The authors refer to a 'Chapter 2', not included in the manuscript.

Experimental design

No comments.

Validity of the findings

The figures are hard to interpret due to their size and color usage. Bigger figures would be advisable, and also the use of different color for each developmental stage. Some data sets seem more sigmoidal than linear (for instance, L1, L2, L3f in Figure 1), whereas others are hardly linear (L3f, P in Figure 2). It would be desirable that the authors include linear regression coefficients for each condition in all their analyses.
It would be important that authors elaborate on how this experimental model, with controlled/constant temperatures, could be applied as a reference for postmortem interval determination. What is the influence of temperature fluctuations (as would happen in a real PMI estimation) on development timing and error rate estimation? This should be discussed more deeply.

Reviewer 2 ·

Basic reporting

This work represents a particularly well-done carrion fly development study that clearly merits publication. The authors’ careful examination of the relationship between temperature and development rate will probably have a large impact on the field of forensic entomology. However, I have quite a few editorial comments and one essential change.

MAJOR COMMENTS

1. Expand on what is meant by “statistically valid development models.” Presumably this refers to using such a model for some statistical inference. What inference, and what could go wrong if one were to use a statistically invalid development model?

2. The L. sericata development literature is rather more extensive than Kamal (1958), Ash and Greenberg (1975), Greenberg (1991), Anderson (2000), and Grassberger and Reiter (2001). I don’t suggest that the authors list every paper, but something like “and others” should acknowledge this fact, and given the extent to which geographic variation in development rate is discussed one of the papers by Tarone and colleagues on this topic should be cited.

3. The authors have reason to be proud of their exceptional degree of replication, but the claims that this is essential (i.e. 4 separate growth chambers/temperature) are simply not justified by extensive experience. Even the classic paper by Hurlbert describes how interspersion, in this situation alternating the chambers used for a temperature, is likely adequate. And anyway it’s impossible to not fall short of the mathematical ideal. If one went by pure theory then the chamber assignment would be truly random only if the chambers were randomly drawn from the entire universe of chambers, the fly eggs randomly drawn from the colony, etc. Unless the authors tested for, and found, a chamber effect that would have been enough to change the results they should drop the section on what is “proper.” The size of this experiment speaks for itself.

4. By what criterion was a larva considered to be postfeeding? The only statement about this that I noticed concerned how to distinguish between postfeeding larva and pupa, but the problematic transition is feeding third larval instar to postfeeding third larval instar. Feeding larvae can leave and return to the food, so being off the food seems like an unreliable clue.

5. “Two regression procedures were used. First, to determine the appropriate transition distributions . . .) This probably reflects my ignorance of the relevant statistical models, but what is meant by “appropriate in this context. Why do the authors care so much about how instar transition ages are distributed? What are the implications for subsequent analyses and practical applications.

6. “In contrast, with forensic use of degree day models, the potential inaccuracy associated with including non-linear data in the calculation model introduces systematic error that could easily be significant in using degree days for estimating post mortem intervals.” One reason I have difficulty in understanding this sentence is that the authors don’t go into how one should estimate PMI. Please give one or more examples of what you mean by a significant systematic error in this context.

7. Finally, the authors need to present their results in the fashion most directly relevant to the task of estimating larval age, namely the mix of development states that were observed at each sample age. This would probably best be in an appendix, and the format could be the raw data (the number of individuals in each rearing container observed to be a particular instar at a particular age and temperature) or a summary table showing the proportion of each instar as a function of age and temperature. I consider this to be essential for publication.


MINOR COMMENTS

1. If each rearing container held 20 insects, and each life stage was observed at 5 ages, and each age was replicated 4 times, I calculate 20*5*4=400 measurements per life stage rather than 20.

2. What is the purpose of “(L. sericata)” in the abstract?

3. There are a few places where the word “data” is not treated as the plural form.

4. “In the past insect develop research has focused on agricultural pests and disease vectors, with accuracy levels of . . .” Accuracy in what sense?

5. “yet development rate concepts from agriculture pest development data sets/models are being applied to postmortem interval estimations, and in doing so imply levels of precision much greater than the data allow.” Please cite examples.

6. The error rate referred to in legal standards for forensic science does not mean statistical confidence limits or other aspects of the calculations. It refers to the error rate in practice, i.e. the likelihood that a competent person performing a particular analysis in casework will reach a correct conclusion.

7. Line 88 has an orphan quotation mark.

8. “With insects, inadvertent selection in colonies most frequently occurs in oviposition behavior and in reduced fecundity” should be supported by a citation(s).

9. “The container was then placed randomly in an incubator.” Describe the randomization procedure.

10. “Each life stage (egg-1st stage . . .” These are transitions between instars, not stages. Furthermore, stage and instar are not synonyms. Blow flies have egg, larval, pupal, and adult stages, and three larval instars.

11. What chapter 2?

Experimental design

No comment

Validity of the findings

No comment

Additional comments

No comment

---

## Round 0.2 · accepted · Accept

As evidenced by the reviewers' written statements, the new version of the article is much improved and is now acceptable for publication in PeerJ

Reviewer 1 ·

Basic reporting

The authors have greatly improved the general organisation of the manuscript and have addressed the concerns raised in the previous submission.

Experimental design

No comments.

Validity of the findings

No comments.

Reviewer 2 ·

Basic reporting

No additional comments.

Experimental design

No additional comments.

Validity of the findings

No additional comments.

Additional comments

The authors have adequately responded to my comments on the earlier version of their manuscript.